# A Domain Incremental Continual Learning Benchmark for ICU Time Series Model Transportability

Ryan King, Conrad Krueger, Ethan Veselka, Tianbao Yang, and Bobak J. Mortazavi

*Computer Science & Engineering, Texas A&M University*, College Station, United States

{kingrc15, conradk1234, veselka, tianbao-yang, bobakm}@tamu.edu

*Abstract*—In recent years, machine learning has made significant progress in clinical outcome prediction, demonstrating increasingly accurate results. However, the substantial resources required for hospitals to train these models, such as data collection, labeling, and computational power, limit the feasibility for smaller hospitals to develop their own models. An alternative approach involves transferring a machine learning model trained by a large hospital to smaller hospitals, allowing them to fine-tune the model on their specific patient data.

However, these models are often trained and validated on data from a single hospital, raising concerns about their generalizability to new data. Our research shows that there are notable differences in measurement distributions and frequencies across various regions in the United States. To address this, we propose a benchmark that tests a machine learning model's ability to transfer from a source domain to different regions across the country. This benchmark assesses a model's capacity to learn meaningful information about each new domain while retaining key features from the original domain.

Using this benchmark, we frame the transfer of a machine learning model from one region to another as a domain incremental learning problem. While the task of patient outcome prediction remains the same, the input data distribution varies, necessitating a model that can effectively manage these shifts. We evaluate two popular domain incremental learning methods: data replay, which stores examples from previous data sources for fine-tuning on the current source, and Elastic Weight Consolidation (EWC), a model parameter regularization method that maintains features important for both data sources.

Finally, we propose a new domain incremental learning method that combines EWC and data replay with the ability to adjust the number of updates utilizing data from previous sources. Our results show that this proposed method outperforms EWC and data replay alone. We also highlight specific shortcomings related to model transferability in the clinical setting, underscoring the need for further research and development in this area.

*Index Terms*—EHR Time Series, Transferability, Continual Learning, Domain Incremental Learning

CLINICAL risk prediction models, trained with machine learning techniques, continue in growth and ubiquity [1], [2]. Increasing access to data, from clinical trials [3] to remote health data [4], has led to an explosion of models across various data sources, clinical settings, and medical outcomes. A key limitation, which has partially contributed to the breadth of models designed, is the lack of transportability in these models. Because of privacy and limitations around sharing of medical data, clinical risk prediction models are often trained and validated internally, with minimal to no external validation. As a result, these models, when externally validated, often fall significantly short of reported performance [5]. Techniques are needed, therefore, to facilitate the transport of functioning models, at reported performance levels, across clinical institutions.

One approach to address this issue is to evaluate how well the models transfer. For example, in [6], a model pretrained on data from a large hospital was evaluated on data from smaller hospitals, testing the robustness of the pretraining on new populations. However, we show that significant differences exist between the data collected from hospitals of different regions. Without adjusting these models to their new data sources, these difference can degrade the performance of a transferred model.

Given these differences, clinics may need to personalize models to their specific demographics and patient outcomes. Instead of starting from scratch, clinics can use models pretrained on similar tasks from different sources as a starting point. This approach is known as the domain incremental learning (DIL) setting [7], where different data sources are used to train models on the same tasks. The goal of this training is to adapt a model to a new domain while remembering information from previous data sources. However, simply training on each new source can lead to a phenomenon called catastrophic forgetting [8] in which information from previous sources is forgotten, losing benefits from beginning with a pretrained, working model.

Many methods have been proposed to mitigate catastrophic forgetting in incremental learning tasks. Replay methods [9] use a memory bank of examples from previous sources to retain information. However, in the clinical setting, privacy constraints restrict the transfer of patient information from one clinic to another. Alternatively, Elastic Weight Consolidation (EWC) [10] regularizes the model parameters by penalizing changes to parameters that are important for previously trained sources. This is achieved by adding a regularization term to the loss function that constrains significant parameters from deviating too much from their original values. Both of these methods help to preserve knowledge from earlier sources while allowing the model to learn from new ones, thereby improving the overall performance and stability of the model in incremental learning scenarios.

We propose a DIL benchmark in which a model starts at a large hospital, Beth Israel Deaconess (MIMIC-III), and is

transferred to various regions of smaller hospitals from the eICU dataset. We utilize the in-hospital mortality (IHM), phenotyping, length of stay (LOS), and decompensation benchmarks [11] to train and evaluate our model at each regional source. We show the significance of the distribution shift between the features of each region. We evaluate two existing DIL methods for overcoming catastrophic forgetting on our proposed benchmark: a replay based method, and a weight regularization method using EWC [10]. Finally, we propose a modification to replay and a new method that combines EWC and the modified replay with the ability to adjust the frequency of the updates using data from previous sources. We compare our proposed DIL methods to EWC, data replay, and an additional baseline which uses no DIL methods. In summary our contributions are as follows:

1) We propose a DIL benchmark for clinical time series data which starts by training a model on a large hospital system, MIMIC-III, and transfers to various regions throughout the United States. We provide a detailed description of the distribution of each region along with a discussion of the key differences.
2) We propose a new DIL method which combines replay and EWC, and a modification to data replay with the ability to adjust the frequency of updates utilizing data from previous sources.
3) We utilize four clinical tasks to evaluate models in a variety of settings include binary classification, multi-label classification, and two sequence to sequence tasks. We evaluate four methods on our proposed benchmark: EWC [10], traditional data replay [7], our adjusted data replay, and our proposed combined method.

## I. RELATED WORKS

In this section, we review some methods for overcoming catastrophic forgetting in the DIL setting. In our benchmark, we limit ourselves to model regularization and data replay methods since dynamic expansion methods add additional computational resources that clinics may not have.

### A. Data Replay

Data replay is a common technique used to mitigate catastrophic forgetting in DIL. Replay methods maintain a memory bank of examples from previous tasks and periodically reintroduce them during training on new tasks. This helps to reinforce previously learned knowledge and prevents the model from forgetting earlier information. Rainbow Memory [9] exemplifies this approach by storing a subset of samples from past tasks and replaying them along with new task data. The balance between new and old data during training is crucial to ensuring that the model retains prior knowledge while effectively learning new information.

### B. Model Regularization

Model regularization techniques aim to preserve important parameters associated with previously learned tasks, thereby reducing catastrophic forgetting. EWC [10] is a prominent method in this category. EWC adds a regularization term to the loss function that penalizes significant changes to parameters deemed important for past tasks. This is achieved by computing the Fisher Information Matrix, which identifies crucial parameters, and then constraining these parameters to remain close to their original values during subsequent training. Another regularization approach, Synaptic Intelligence (SI) [12], calculates the importance of each parameter in an online manner and adjusts the loss function to protect these important weights.

### C. Dynamic Expansion

Dynamic expansion methods address the limitations of fixed model architectures by expanding the model as new tasks are introduced. These methods add new neurons or layers to the network, enabling it to adapt to new tasks without interfering with existing knowledge. Progressive Neural Networks [13] illustrate this approach by creating new pathways for each task while retaining the previously learned pathways. This allows the model to leverage prior knowledge without risking interference. Similarly, the Dynamically Expandable Network [14] selectively expands the network by adding neurons only when necessary, based on a sparsity constraint and a splitting criterion. This method ensures that the model remains efficient while being capable of learning new tasks incrementally.

## II. METHODS

In the following sections, we go over our data extraction and preprocessing as well as our model training and evaluation methods. Data preprocessing is essential to performance as both the distribution of data and the characteristics of tasks strongly affect outcome. In our approach we use the same preprocessing methods and benchmark standard BiLSTM and LSTM models as in [11]. Throughout this section we will be training, validating, and testing a model on dataset $\mathcal{D} = \{D^1, D^2\}$ where $D^1 \cap D^2 = \emptyset$ and $N_t = |D^t|$.

### A. Data

The MIMIC-III dataset is a collection of clinical measurements (vitals, lab results, demographics, nursing reports, diagnoses, length of stay, mortality) from over 50,000 patient stays during hospital admission between 2001 and 2012 at the Beth Israel Deaconess Medical Center [15]. The eICU dataset is a collection of similar clinical measurements from over 200,000 patient stays across multiple centers in the United States [16]. The eICU dataset timestamps these measurements in minutes after ICU admission [16]. We use the 17 measurements described in the benchmark tasks [11]: heart rate, mean arterial pressure (MAP), diastolic blood pressure (DBP), systolic blood pressure (SBP), oxygen saturation, respiratory rate (RR), temperature in Celcius, glucose, fraction of inspired oxygen (FiO2), pH, height, weight, Glasgow coma score total, Glasgow coma score eyes, Glasgow coma score motor, Glasgow coma score verbal, and capillary refill.

Multiple exclusion criteria were applied before training. Patients younger than 18 were excluded due to the significant

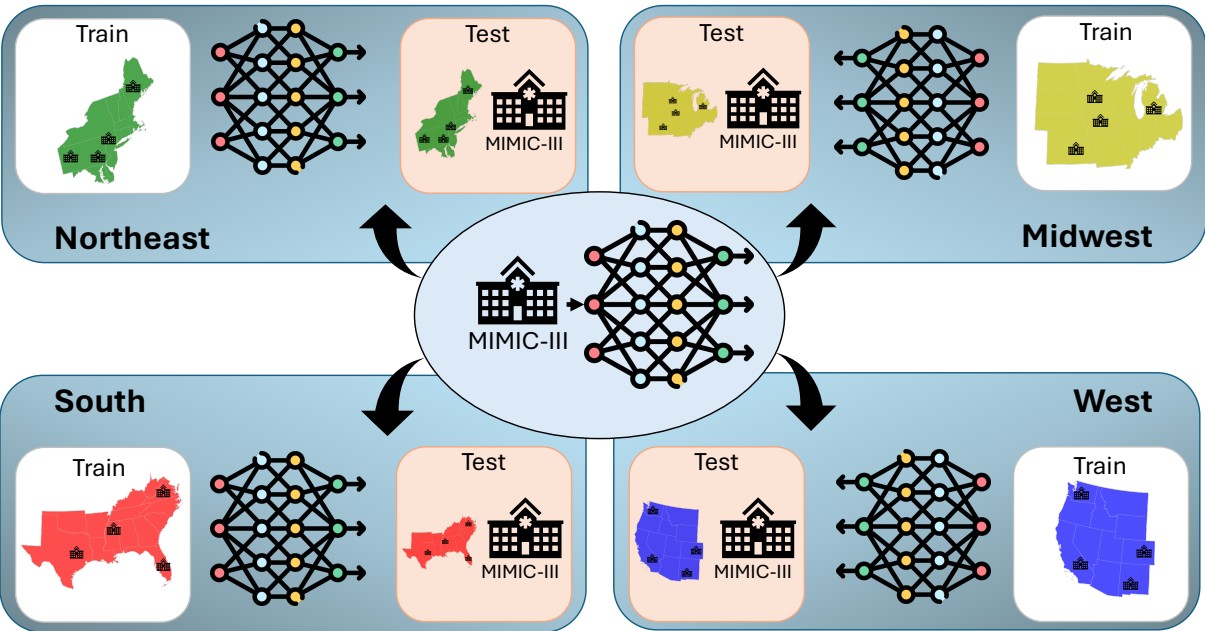

Fig. 1. We illustrate our proposed benchmark frame work. A model is initialized on the MIMIC-III dataset and trained on a clinical outcome prediction task. The model is then transferred to each of the four regions in the United States and trained on the same task. During evaluation, the model is then evaluated on a held out set of data from the MIMIC-III dataset and the respective region.

biological differences between pediatric and adult patients [11]. Patients with 2 or more ICU stays or were transferred in a hospital admission were excluded [11]. This means patients who were admitted to the ICU multiple times, but during distinct hospital admissions, were included. Inspired by [17], we eliminated patients who had less than 15 records in the ICU. For in-hospital-mortality, patients who were in the ICU for less than 48 hours were excluded [11]. For length-of-stay and decompensation, a minimum length-of-stay of 5 hours was applied to minimize the risk of models seeing too little data to make an informed prediction [11]. We also ensured no task had any empty samples, unknown length of stay, data prior to ICU admission, or data after ICU discharge [11]. For mortality prediction tasks, we eliminated admissions with inconsistent labels between hospital and unit discharge status. For example, a patient reported as being expired after ICU discharge but alive after leaving the hospital would be excluded.

### B. Data Preprocessing

We applied the first set of exclusion criteria by filtering the patient unit stay ID that had a known age above or equal to 18 and was the only patient unit stay ID underneath it's corresponding patient health system ID, which represents a single hospital stay. We also checked that their maximum unit visit number was 1. We unified measurements that were the same but differently labeled into the same column [17]. For example, there are invasive and non-invasive systolic blood pressure records taken by nurses. Since, these are measuring the same vital (systolic blood pressure), we grouped these readings together. All categorical variables, besides capillary refill, were already numerically encoded, so no additional preprocessing had to be done. For capillary refill, we encoded

readings of "normal" and "< 2 seconds" as normal (0) and "> 2 seconds" as abnormal (1). We also only included admission height and weight once at timestamp zero, since there was no official timestamp of when the measurement was taken. We noticed MAP had missing regional information in the Northeast and the West. To address this, we decided to unify MAP with non-invasive and invasive mean blood pressure readings. We then took the union of timestamp measurements for each feature per patient unit stay ID [17].

When data is prepared to be loaded into batches for training, another set of preprocessing steps is applied. To make the time series data have consistent interval readings across all patients, the data is binned into hourly observations, where the last non-null value in the bin is used [11]. In the instance where a bin contains no observations, bin values are imputed using previously recorded values [11]. If no value can be imputed via forward filling, then a normal value from a predefined table is utilized [11].

Lastly, an imputation indicator is applied. An additional column for every feature is created to have a binary label indicating whether or not a feature measurement was imputed [11]. That is, showing if the number in a particular bin is a true measurement or filled via forward filling or a normal value [11]. Categorical features are one hot encoded and the 17 additional columns for imputation indication are all combined, leading to 76 columns in total [11].

After processing patient stays are organized into the regional splits shown in figure 1. Each patient stay within the eICU dataset is registered to a specific hospital, which are grouped by region into 4 categories: South, Midwest, West, and Northeast; hospitals with no region label (approximately 6% of patient stays) are excluded.

### C. Benchmark Tasks

We group training and testing sets by region for the four benchmark prediction tasks including in-hospital-mortality, decompensation, length-of-stay, and phenotyping from [11]. The continual learning problem is formulated as the sequential training of the model from the MIMIC-III data initially to one of the four eICU regions for the respective benchmark task.

*1) In-hospital mortality prediction:* The in-hospital mortality (IHM) benchmark is a binary classification task using the first 48 hours of the ICU stay to predict patient mortality, evaluated using the area under the receiver operating characteristic AUC-ROC as the primary metric, and area under the precision-recall curve (AUC-PR) as the secondary metric.

*2) Decompensation prediction:* Similarly to IHM, the decompensation benchmark is a binary classification task with the goal of predicting mortality, but now predicting patient decompensation within the next 24 hours for each hour of an ICU stay. The metrics are likewise AUC-ROC and AUC-PR.

*3) Length-of-stay prediction:* The length-of-stay (LOS) benchmark is a 10 class classification task that aims to predict the remaining duration of stay in the ICU for a patient at each hour of the stay. The 10 classes distinguish a less than 1 day stay, seven day-long classes for each of the days of the first week, one for over a week but less than two, and the final class for over two weeks. The primary metric used for evaluation is the Cohen Kappa score, with the mean absolute deviation (MAD) as the secondary metric.

*4) Phenotype classification:* The phenotyping benchmark involves the classification of the 25 acute care conditions described in Table 3, using a one-versus-rest strategy this becomes a multi-label binary classification task, and is evaluated using a macro-averaged AUC-ROC as the primary metric, and micro-averaged AUC-ROC as the secondary metric.

### D. Model Architecture

We use the standard BiLSTM model from [11] for IHM and Phenotyping tasks, consisting of 76 input features, the specified number of BiLSTM layers (2 for IHM, 1 for the rest), and the specified number of hidden dimensions. For LOS and Decompensation, we utilize an LSTM model similar to [11]. However, this LSTM is not bi-directional. Dropout is applied if the number of layers is greater than 1. This forms the LSTM layer, which is followed by a final linear layer that applies dropout accordingly and returns the sigmoid activation of the output for the specified number of classes.

### E. Model Training

For each of the four benchmark tasks, training iterations begin by initializing the model and loading in validation and testing data for MIMIC-III and the specified eICU region. The model is first trained on the MIMIC-III data, then on the specified eICU region. The model is evaluated on both source's validation sets at the end of each training epoch, and tested on both test sets after training on the target source is finished. PyTorch's model eval mode is used during both evaluation and testing so model weights cannot change. When a target source is finished training, a memory buffer with a predefined capacity is updated with random training samples from that source; in the case of more than two sequential sources (not shown here), the buffer maintains an equal number of samples from each previous source that sum to the specified capacity. The memory buffer is used to compute both the EWC regularization term and the replay loss.

For IHM, Decompensation, and Phenotyping, the current target training loss (binary cross entropy/BCE) is defined as:

$$\mathcal{L}_{curr} = -\frac{1}{N_t} \sum_i^{N_t} (y_i \cdot \log \hat{y}_i + (1 - y_i) \cdot \log(1 - \hat{y}_i)))$$

and for LOS (cross entropy/CE) is defined as:

$$\mathcal{L}_{curr} = -\frac{1}{N_t} \sum_i^{N_t} \sum_j^C (y_{ij} \cdot \log \hat{y_{ij}} + (1 - y_{ij}) \cdot \log(1 - \hat{y_{ij}}))$$

where $N_t$ is the total number of training samples for the source, $y$ is the ground truth label, $\hat{y}$ is the predicted label, and $C$ is the number of classes. The Adam optimizer [18] is used for all training tasks.

*1) Replay:* Replay loss is traditionally calculated as

$$\mathcal{L} = \frac{1}{s} \cdot \mathcal{L}_{curr} + \left(1 - \frac{1}{s}\right) \cdot \mathcal{L}_{rep}$$

where $s$ is the number of sources seen so far, and $\mathcal{L}_{rep}$ is the appropriate loss (BCE or CE) on random samples from the memory buffer for the respective benchmark task; the model trains on a random sample from the memory buffer for each sample trained from the current source and the respective losses are weighted proportionally to the number of sources seen so far [7]. This method can be vulnerable to overfitting when the memory buffer is not representative of the training data (i.e the buffer is small) [19], [20] so we adjust replay and test it as described in the following section.

*2) EWC:* EWC aims to maintain model weights that are important to prediction on previous sources (elasticity) without severely limiting learning on new sources [10]. This is achieved via a quadratic penalty term approximated using the diagonal precision given by the diagonal of the Fisher Information Matrix $F$ to determine the relative importance of parameters to previous sources and regulate changes to those parameters. This penalty is weighted by an importance $\lambda$ and concatenated with the Loss on the current source:

$$\mathcal{L}_{ewc} = \mathcal{L}_{curr} + \lambda \sum_i \frac{1}{2} F_i (\theta_i - \theta_{A,i}^*)^2$$

$$F_i = \frac{1}{N_b} \left(\frac{\partial \mathcal{L}_{rep}}{\partial \theta_i}\right)^2$$

where $\theta_i$ represents each parameter from the current source, $\theta_{A,i}^*$ represents parameters from the previous source(s), and $\mathcal{L}_{rep}$ is $\mathcal{L}_{curr}$ calculated on the replay buffer with size $N_b$. In EWC all samples from the memory buffer are used to calculate the penalty. The diagonal of $F$ is approximated via the normalized sum of squares of the first partial derivative of the loss on the memory buffer with respect to $\theta_i$ for each $F_i$.

However, these approaches must balance the memory cost of a sufficiently representative buffer against the risk of overfitting to a subset of the training data. This poses a significant challenge with the small or non-existent memory necessitated in health care settings due to limited transferability of patient data between ICU's. For this reason, we propose an adjustment to the replay implementation to minimize risk of overfitting in two primary ways:

$$\mathcal{L}_{adj} = \frac{1}{N} \sum_{i}^{N} \left( \left( 1 - \frac{1}{s} \right) \cdot \mathcal{L}_{curr} + \frac{1}{s} \cdot \mathcal{L}_{rep,j(i)} \right)$$

Where

$$\mathcal{L}_{comb} = \begin{cases} \mathcal{L}_{adj} & \text{if } i \bmod p = 0 \\ \mathcal{L}_{curr} & \text{if } i \bmod p \neq 0 \end{cases}$$

and $p = \left\lfloor \frac{N}{\text{Buffer Size}} \right\rfloor$ and $j(i) = \lfloor i/p \rfloor$. Here $\mathcal{L}_{rep,j(i)}$ is $\mathcal{L}_{curr}$ calculated on data from the memory buffer at the index defined by $j(i)$. $N_t$ is the total number of samples, $s$ again specifies the number of sources seen so far, and $i$ represents the sample index. In our proposed combined method $\mathcal{L}_{curr}$ is replaced by $\mathcal{L}_{ewc}$ in both $\mathcal{L}_{adj}$ and $\mathcal{L}_{comb}$ above. Note: Buffer Size is always less than $N_t$, so $p > 1$.

We first reverse the progressive weighting on the replayed samples to discourage overfitting to the small subset of data in the memory buffer while training later sources. Secondly, replay is performed and the loss adjusted at even intervals during training such that each sample within the buffer is seen at most once, in effect acting as a proportionate set of additional training samples. This is achieved by adjusting the loss periodically based on the value of $p$ and incrementing the sample index $j(i)$ from which the replay loss is calculated only at each adjustment. If $p = 2$, the loss is adjusted every 2 steps starting from $i = 0$ where $j(0) = 0$, then the second adjustment $j(2) = 1$, and so on.

We have observed that our adjusted replay can improve over its traditional implementation with limited memory, especially in the case of multiple sequential sources where the buffer remains limited in size, though in our testing only 2 sequential sources are shown. Our combined method often outperforms applications of these methods individually, though this difference is not as significant. We expect the performance gap between our combined method and traditional DIL methods to grow with the number of sources.

| Task | Buffer Size | Samples | Importance | Epochs |
|---|---|---|---|---|
| IHM | 500 | All | 6 | 4 |
| Phenotyping | 500 | All | 4 | 6 |
| Decompensation | 3500 | 100k | 6 | 1 |
| LOS | 3500 | 100k | 6 | 1 |

TABLE I
ADDITIONAL HYPERPARAMETERS.

*F. Model evaluation, hyperparameter tuning, and validation*

The model's performance on all sources is evaluated at the end of training on each source as described in the previous section. For our 2 source setup, this results in scores for the eICU region after training only on MIMIC-III, effectively describing the forecasting capability of the model. After training on the eICU region, the model is tested again on both MIMIC-III and the eICU region.

We evaluate performance utilizing a per-source average (PSA) value, calculated after training on each source, that is the average of the performance $P$ on all sources the model is currently trained on: $PSA = \frac{1}{s} \sum_{i}^{s} (P_i)$ where $i$ is the source number and $s$ is the total number of sources seen so far. An ideal (high) PSA is achieved when a model maximizes learning on new sources while minimizing forgetting on old sources. Results reported in Table IV show the mean and standard deviations of five complete training iterations in which a model is trained and evaluated on MIMIC-III and then the given eICU subregion. We outline in Table I the parameters we found for optimal performance on each of the 4 clinical prediction tasks: Importance defines the weight on the EWC regularization penalty, and is 0 in the absence of EWC. We maintain the model parameters as in [11], and tune importance and the number of training epochs (excluding buffer size and sample size which remain constant) via grid search on the validation sets. The search space was limited to even values between 1 and 10 epochs for IHM and Phenotyping, and importance was limited to even values between 1 and 8 for all tasks.

For IHM and phenotyping the epochs parameter describes how many times the model trains on each source's entire dataset before moving on to the next source. In the case of Decompensation and LOS there are far more samples due to the sequence to sequence nature of the tasks, and training on more than a small subset can easily result in overfitting. For this reason we set epochs to 1, and we train on a subset of the data with source sample sizes according to approximate region split ratios, adjusting sample size accordingly from a basis of 100-thousand at which consistent benchmark performance on the MIMIC-III split is achieved, resulting in 100-thousand, 100-thousand, 50-thousand, and 25-thousand samples respectively for the South, Midwest, West, and Northeast eICU regions.

## III. EXPERIMENTS AND RESULTS

In this section, we provide an analysis of our proposed benchmark dataset. We highlight some of the differences between each task in our datasets and discuss how these shifts can cause issues when transferring to new domains. We then evaluate EWC and our modified replay on our proposed benchmark along with our proposed method. Code for our experiments, along with our data preprocessing pipeline, are available on github at https://github.com/kingrc15/EHRTransferBenchmark

*A. Dataset*

In this section we analyze each task in our proposed dataset and highlight some of the key differences. We start by looking at the distribution of the features in the dataset. From there we look at the frequency of both the measurements and the labels for each region and benchmark task.

*1) Measurement Distributions:* We report the results in Figure 2. The distribution for each measurement was calculated

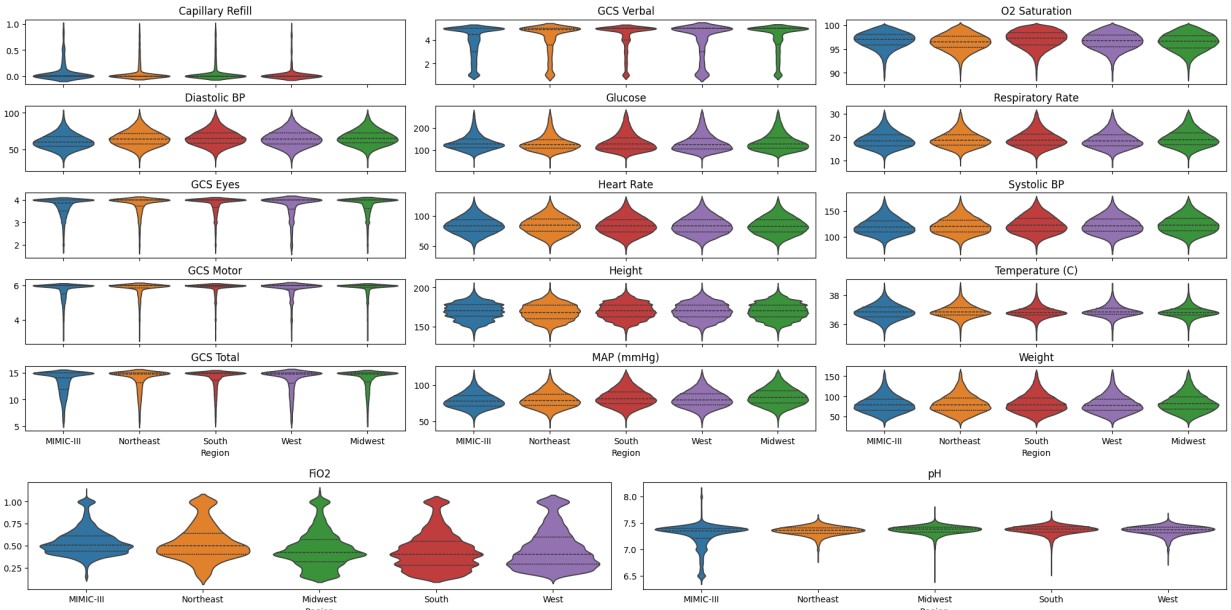

Fig. 2. We plot the distribution of each measurement in our dataset across each of our tasks. In each plot, we see each region along with a dashed lined indicating the mean of the distribution and a dotted line indicating the inner quartile range. Plots with missing distributions indicate that the measurement was not taken in that region.

| Region | MIMIC-III | Midwest | South | West | Northeast |
|---|---|---|---|---|---|
| Capillary | 0.0025 | 0 | 0.0237 | 0.0110 | 0.1200 |
| DBP | 1.0641 | 1.1045 | 0.9663 | 1.4268 | 1.0844 |
| FiO2 | 0.0597 | 0.0207 | 0.0227 | 0.0219 | 0.0144 |
| GCS Eyes | 0.1143 | 0.1091 | 0.2287 | 0.1051 | 0.5027 |
| GCS Motor | 0.1238 | 0.1088 | 0.2286 | 0.1050 | 0.5026 |
| GCS Verbal | 0.1242 | 0.1750 | 0.2359 | 0.1562 | 0.5027 |
| GCS Total | 0.1615 | 0.1085 | 0.2253 | 0.1050 | 0.5027 |
| Glucose | 0.2462 | 0.2061 | 0.1819 | 0.1824 | 0.2540 |
| HR | 1.1532 | 1.2787 | 1.1086 | 1.5929 | 1.0707 |
| Height | 0.0049 | 0.0267 | 0.0266 | 0.0319 | 0.0263 |
| MAP | 1.0563 | 1.0795 | 0.9931 | 1.3693 | 1.0376 |
| O2 Sat | 1.1260 | 1.1963 | 0.9953 | 0.8140 | 0.7383 |
| RR | 1.1519 | 1.1638 | 1.0612 | 1.2426 | 1.0300 |
| SBP | 1.0646 | 1.1044 | 0.0966 | 1.4268 | 1.0845 |
| Temperature | 0.3198 | 0.3366 | 0.3092 | 0.4444 | 0.3391 |
| Weight | 0.0427 | 0.0255 | 0.0269 | 0.0323 | 0.0263 |
| pH | 0.0977 | 0.0233 | 0.0219 | 0.0180 | 0.0228 |

TABLE II
WE REPORT THE AVERAGE FREQUENCY OF EACH MEASUREMENT FOR EACH REGION.

by plotting the frequency of average measurement of every patient episode.

We see several differences between the distributions of each region. Notably, we observe a large change in FiO2 and pH distribution between MIMIC and each of the eICU regions. We also note that capillary refill has missing regional information. This indicates that in certain regions, the measurements we are concerned with simply are not recorded. This leads us to investigate the frequency of each of the measurements.

*2) Measurement Frequencies:* We measure the distributions of each regional split of the eICU dataset [16] and the MIMIC-III dataset [15]. Table II indicates the frequency of

a measurement occurring during a patient's stay.

$$F = \frac{1}{n} \sum_{i=0}^{n} c_i / t_i$$

where $n$ is the number of patient stays in the cohort, $c_i$ is the number of occurrences of a particular measurement of the $i$th patient stay, and $t_i$ is the length-of-stay of patient stay $i$ in hours. Thus, a higher frequency means the measurement was more often recorded during a patient's ICU stay.

An important observation is there are changes in frequency of readings in different regions of the United States. For Glasgow Coma Scores, Table II indicates that doctors and nurses in the northeast are more likely to measure these scores than in the south. There are also differences in measurement frequencies between the two datasets. For example, FiO2 is around 0.06 in MIMIC-III but between 0.015 to 0.025 across all eICU regions. Capillary Refill is so infrequently recorded that they show zero recordings in the Midwest.

In Figure 2, the shape of the distributions for each region and MIMIC-III is shown. From here we can see that most measurements, regardless of their frequency, have a similarly shaped distribution and nearly identical means. Some measurements may have a thicker or wider distribution as a result of a larger data pool or recorded range of measurements. The most notable distribution variations are in pH and FiO2 as mentioned before. MIMIC-III seems to have a larger range compared to any of the regions in eICU. MIMIC-III also appears to have a thinner tail near the 0.25 mark for FiO2. However, the eICU dataset shows a larger concentration of readings under 0.25.

This emphasizes the unique characteristics and resulting differences between datasets and regions of the United States.

| Label | MIMIC-III | South | Midwest | West | Northeast |
|---|---|---|---|---|---|
| AURF | 0.2139 | 0.1130 | 0.1055 | 0.0996 | 0.1935 |
| ACD | 0.0735 | 0.0705 | 0.0705 | 0.0823 | 0.0647 |
| AMI | 0.1035 | 0.0626 | 0.0497 | 0.0516 | 0.0637 |
| CD | 0.3212 | 0.1300 | 0.0988 | 0.1138 | 0.2483 |
| CKD | 0.1338 | 0.0952 | 0.0901 | 0.0777 | 0.1081 |
| COPD | 0.1302 | 0.0810 | 0.0717 | 0.0712 | 0.1187 |
| CS | 0.2075 | 0.0086 | 0.0075 | 0.0094 | 0.0141 |
| CoDi | 0.0719 | 0.0097 | 0.0074 | 0.0063 | 0.0127 |
| CHF | 0.2678 | 0.1070 | 0.0739 | 0.0869 | 0.1219 |
| CA | 0.3231 | 0.0436 | 0.0210 | 0.0140 | 0.0376 |
| DMC | 0.0952 | 0.0457 | 0.0337 | 0.0370 | 0.0429 |
| DM | 0.1927 | 0.0070 | 0.0031 | 0.0020 | 0.0108 |
| LD | 0.2902 | 0.0603 | 0.0075 | 0.0306 | 0.0792 |
| EH | 0.4194 | 0.1854 | 0.0770 | 0.1587 | 0.2011 |
| FD | 0.2686 | 0.1196 | 0.0828 | 0.0968 | 0.3463 |
| GH | 0.0732 | 0.0586 | 0.0511 | 0.0573 | 0.0743 |
| HWC | 0.1324 | 0.0181 | 0.0091 | 0.0121 | 0.0164 |
| OLD | 0.0889 | 0.0271 | 0.0254 | 0.0329 | 0.0720 |
| LR | 0.0517 | 0.0273 | 0.0224 | 0.0212 | 0.0445 |
| UR | 0.0406 | 0.0047 | 0.0047 | 0.0050 | 0.0077 |
| Pleurisy | 0.0873 | 0.0282 | 0.0200 | 0.0251 | 0.1221 |
| Pneumonia | 0.1388 | 0.0915 | 0.0948 | 0.0912 | 0.1709 |
| RF | 0.1806 | 0.2109 | 0.1743 | 0.2182 | 0.2958 |
| Septicemia | 0.1426 | 0.0920 | 0.1110 | 0.1702 | 0.1529 |
| Shock | 0.0785 | 0.0504 | 0.0428 | 0.0741 | 0.1163 |
| IHM | 0.1323 | 0.1149 | 0.0854 | 0.1438 | 0.1359 |
| Decomp | 0.0206 | 0.0178 | 0.0139 | 0.0234 | 0.0243 |
| RLoS | 135.39 | 106.247 | 105.379 | 167.528 | 113.196 |

TABLE III

WE REPORT THE AVERAGE GROUND TRUTH VALUE OF EACH REGION FOR EACH TASK. AURF IS ACUTE AND UNSPECIFIED RENAL FAILURE, ACD IS ACUTE CEREBROVASCULAR DISEASE, AMI IS ACUTE MYOCARDIAL INFARCTION, CD IS CARDIAC DYSRHYTHMIAS, CKD IS CHRONIC KIDNEY DISEASE, COPD IS CHRONIC OBSTRUCTIVE PULMONARY DISEASE, CS IS COMPLICATIONS OF SURGICAL/MEDICAL CARE, CoDi IS CONDUCTION DISORDERS, CHF IS CONGESTIVE HEART FAILURE, CA IS CORONARY ATHEROSCLEROSIS, DMC IS DIABETES MELLITUS WITH COMPLICATIONS, DM IS DIABETES MELLITUS WITHOUT COMPLICATION, LD IS LIPID DISORDERS, EH IS ESSENTIAL HYPERTENSION, FD IS FLUID DISORDERS, GH IS GASTROINTESTINAL HEMORRHAGE, HWC IS HYPERTENSION WITH COMPLICATIONS, OLD IS OTHER LIVER DISEASES, LR IS LOWER RESPIRATORY DISEASE, UR IS UPPER RESPIRATORY DISEASE, RF IS RESPIRATORY FAILURE, IHM IS IN-HOSPITAL-MORTALITY, DECOMP IS DECOMPENSATION, AND RLoS IS REMAINING LENGTH OF STAY

There can be varying patient demographics among hospitals, and hospitals may prioritize recording certain measurements.

### B. Benchmark

With our proposed dataset, we evaluate three different DIL methods: EWC, our modified data replay, and our proposed method. This section starts by outlining the experiment setup. We then present the results of our experiments.

*1) Experiment Setup:* The original eICU benchmark [17] performs a standard 5-fold cross-validation across unique ICU visits for each benchmark test. The MIMIC-III benchmark [11] task performs a 70–15-15 train-validation-test split across unique ICU visits for each benchmark test. The validation split was used for hyperparameter tuning and model selection [11]. We similarly use a 70-15-15 train-test split. To prevent data leakage, all patients with more than one ICU visit were included in the same split (both regional split and train-validation-test split). Here it is important that the test set acts as a representative subset of the training data. The proportions

of the ground truth values for each benchmark task across the splits are seen in Table III.

We run tests for the baseline performance of the model, as well as each of the four aforementioned methods: EWC, replay, our adjusted replay, and our proposed combined method, for each of the 4 eICU regional splits after first training the model on the MIMIC-III region. To get baseline results, no EWC or replay based method is used and standard training occurs for each region, allowing comparison for the forgetting prevented by transfer learning methods.

*2) Results:* Table IV summarizes the per-source average performance for each benchmark task, method, and region. For MIMIC-III there is no specification of method, as performance on the first source is unaffected by any applied DIL method. The values shown for MIMIC-III are the average performance of the model across all tests, and approximate benchmark performance of the BiLSTM and LSTM in [11] replicated for each benchmark task.

For both IHM and Phenotyping, our proposed combined method demonstrates equal or greater performance over all other methods for every eICU region. Baseline performance on IHM is already relatively high, in contrast to Phenotyping where improvement is more significant. This is due in part to IHM being the simplest of all tasks, and because there is strong correlation in IHM between MIMIC-III and the eICU regions. We see similar behavior in Decompensation for the Northeast region, where the baseline outperforms all other methods due to high correlation with MIMIC-III, though EWC is close. The optimal method changes with each remaining region in Decompensation, and we observe a significant drop in performance going from MIMIC-III to any eICU region for the LOS task despite the large relative improvement over baseline, highlighting the difficulty of these sequence to sequence tasks.

### IV. LIMITATIONS AND FUTURE DIRECTIONS

We develop the benchmark in this paper to highlight the challenges present when transferring from one clinic to the next. We evaluate two existing methods and an alternative method which we proposed. However, each of these methods requires some patient data be passed from one clinic to the next. Future work could focus on memory-less methods for transferring from one domain to the next.

### V. DISCUSSION AND CONCLUSION

In this paper, our study addresses the critical challenge of deploying machine learning models for clinical outcome prediction in smaller hospitals, which often lack the resources to develop their own models. By proposing a benchmark to evaluate the transferability of models trained in large hospitals to different regions, we emphasize the importance of assessing generalizability across diverse patient populations. Our research highlights significant regional differences in measurement distributions and frequencies, underscoring the need for models that can adapt to these variations.

We conceptualize this transfer as a DIL problem, maintaining consistent prediction tasks while accommodating variations in input data distributions. Our evaluation of two

| | Region | In-Hospital Mortality | | Phenotyping | | Decompensation | | LOS | |
|---|---|---|---|---|---|---|---|---|---|
| | | AUC-ROC | AUC-PR | Macro | Micro | AUC-ROC | AUC-PR | Kappa | MAD |
| | MIMIC-III | 0.836 (0.002) | 0.466 (0.006) | 0.763 (0.001) | 0.815 (0.001) | 0.867 (0.008) | 0.225 (0.006) | 0.331 (0.015) | 0.727 (0.005) |
| Baseline | | 0.848 (0.007) | 0.520 (0.015) | 0.662 (0.003) | 0.709 (0.005) | 0.815 (0.042) | 0.208 (0.038) | 0.068 (0.023) | 0.700 (0.007) |
| EWC | | 0.855 (0.007) | 0.545 (0.012) | 0.662 (0.004) | 0.712 (0.006) | 0.844 (0.006) | 0.241 (0.020) | 0.148 (0.046) | 0.706 (0.009) |
| Replay | South | 0.864 (0.005) | 0.555 (0.007) | 0.728 (0.002) | 0.799 (0.002) | 0.836 (0.021) | 0.213 (0.013) | 0.199 (0.017) | 0.711 (0.012) |
| Adj Replay | | 0.864 (0.004) | 0.569 (0.010) | 0.741 (0.002) | 0.809 (0.002) | 0.797 (0.032) | 0.207 (0.025) | 0.178 (0.038) | 0.714 (0.008) |
| Combined | | **0.864 (0.006)** | 0.565 (0.009) | **0.744 (0.002)** | 0.811 (0.002) | **0.844 (0.010)** | 0.218 (0.027) | **0.200 (0.013)** | 0.705 (0.016) |
| Baseline | | 0.847 (0.007) | 0.505 (0.006) | 0.661 (0.006) | 0.711 (0.005) | 0.768 (0.027) | 0.186 (0.014) | 0.095 (0.018) | 0.707 (0.014) |
| EWC | | 0.846 (0.007) | 0.510 (0.006) | 0.658 (0.006) | 0.714 (0.008) | 0.753 (0.060) | 0.161 (0.032) | 0.198 (0.042) | 0.698 (0.005) |
| Replay | Midwest | 0.855 (0.008) | 0.528 (0.009) | 0.720 (0.004) | 0.802 (0.002) | 0.777 (0.018) | 0.183 (0.013) | **0.230 (0.018)** | 0.706 (0.008) |
| Adj Replay | | 0.860 (0.006) | 0.541 (0.008) | 0.731 (0.004) | 0.805 (0.004) | 0.744 (0.016) | 0.168 (0.009) | 0.200 (0.026) | 0.709 (0.011) |
| Combined | | **0.863 (0.005)** | 0.540 (0.008) | **0.731 (0.003)** | 0.807 (0.002) | **0.793 (0.011)** | 0.183 (0.008) | 0.227 (0.018) | 0.703 (0.010) |
| Baseline | | 0.846 (0.005) | 0.551 (0.011) | 0.669 (0.009) | 0.721 (0.007) | 0.800 (0.007) | 0.188 (0.011) | 0.036 (0.049) | 0.731 (0.010) |
| EWC | | 0.847 (0.007) | 0.550 (0.013) | 0.679 (0.010) | 0.732 (0.011) | **0.806 (0.010)** | 0.202 (0.012) | 0.149 (0.017) | 0.722 (0.007) |
| Replay | West | 0.859 (0.003) | 0.577 (0.012) | 0.725 (0.002) | 0.806 (0.001) | 0.789 (0.009) | 0.198 (0.020) | 0.176 (0.009) | 0.725 (0.011) |
| Adj Replay | | 0.859 (0.007) | 0.576 (0.010) | 0.728 (0.003) | 0.806 (0.003) | 0.787 (0.008) | 0.194 (0.006) | 0.158 (0.041) | 0.715 (0.013) |
| Combined | | **0.860 (0.004)** | 0.578 (0.010) | **0.732 (0.004)** | 0.808 (0.003) | 0.797 (0.014) | 0.215 (0.006) | **0.177 (0.019)** | 0.719 (0.018) |
| Baseline | | 0.853 (0.007) | 0.569 (0.010) | 0.677 (0.007) | 0.731 (0.004) | 0.864 (0.006) | 0.236 (0.007) | 0.072 (0.052) | 0.640 (0.011) |
| EWC | | 0.863 (0.007) | 0.578 (0.015) | 0.679 (0.008) | 0.733 (0.010) | **0.865 (0.006)** | 0.234 (0.014) | 0.144 (0.021) | 0.642 (0.009) |
| Replay | Northeast | 0.874 (0.003) | 0.606 (0.008) | 0.718 (0.003) | 0.810 (0.002) | 0.839 (0.012) | 0.210 (0.015) | 0.170 (0.008) | 0.650 (0.009) |
| Adj Replay | | 0.874 (0.005) | 0.603 (0.011) | 0.720 (0.006) | 0.808 (0.005) | 0.860 (0.012) | 0.220 (0.010) | 0.178 (0.009) | 0.659 (0.008) |
| Combined | | **0.877 (0.003)** | 0.601 (0.008) | **0.722 (0.006)** | 0.810 (0.002) | 0.850 (0.015) | 0.218 (0.020) | **0.180 (0.008)** | 0.647 (0.010) |

TABLE IV

HERE WE DISPLAY THE PER-SOURCE AVERAGE PERFORMANCE OF EACH METHOD ACROSS THE 4 CLINICAL PREDICTION TASKS.

continual learning methods, data replay and EWC, demonstrates their effectiveness and limitations. Building on these findings, we introduce a method that combines EWC and data replay, enhancing model performance by adjusting the number updates with data from previous sources.

Our proposed method shows superior performance compared to using EWC and data replay independently for the non-sequence to sequence tasks. However, this study also reveals specific challenges in model transferability within the clinical setting, indicating areas where further research and innovation are necessary. Ultimately, our work aims to facilitate the broader adoption of robust machine learning models in smaller hospitals, improving clinical outcomes through more accessible and adaptable predictive tools.

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
