# OpenReview forum: "A Domain Incremental Continual Learning Benchmark for ICU Time Series Model Transportability"
_IEEE.org/EMBS/BHI/2024/Conference — IEEE BHI'24_

### Official Review · Reviewer_icfD · 2024-07-25
**A Domain Incremental Continual Learning Benchmark for ICU Time Series Model Transportability**

**Overall Rating:** 2
**Confidence:** 5

**Other Quality Metrics:**

(a) Clarity of writing: poor
(b) Clinical Significance: great
(c) Methodological Novelty: fair
(d) Experiments and Results: fair

**Questions For The Authors:**

1. You tested a replay-based method in your benchmark, which stores examples from previous tasks. However, patient information is private and cannot be shared among hospitals. Why, then, can you store actual information from the MIMIC dataset in the model and adapt it to the eICU dataset?
2. On Page 2, you mention, "in our case, T is simply 2." This is confusing because you previously indicated that training, validation, and testing are three separate sets.
3. On Page 3, you state, "Multiple exclusion criteria were applied before training." This would benefit from a figure showing the number of samples excluded at each step and the final statistical results of your data.
4. On Page 3, the paragraph "Lastly, an imputation indicator is applied..." is unclear. Please use formulas, examples, or figures to clarify this explanation.

**Strengths:**

The topic is highly relevant as it addresses actual challenges faced by hospitals.

**Summary Of The Paper:**

The paper introduces a benchmark designed to assess the ability of machine learning models to transfer clinical outcome prediction capabilities from one hospital to multiple regions across the United States. It primarily addresses the challenge of domain incremental learning within a clinical context. The study evaluates established domain incremental learning methods and proposes a novel method that integrates these approaches.

**Weaknesses:**

1. Numerous typographical and formula errors hinder the reviewer's understanding of the paper. For example, on Page 4, two instances of $\mathcal{L}_{curr}$ are defined, but no formulas for $\mathcal{L}_{rep}$ are provided. On Page 5, $\mathcal{L}_{comb}$ does not correspond to the formula above it. Typographical errors include, but are not limited to, the repeated phrase "we will be be training" on Page 2.
2. The model training process is unclear. The authors state, "The model is initially trained on the MIMIC-III data and evaluated on the validation sets of both MIMIC-III and the specified eICU region during each epoch" on Page 4. However, based on the research topic, a model should not access both datasets simultaneously.
3. The evaluation metrics are inadequate. For modality and disease detection, essential clinical metrics such as sensitivity and specificity are missing.

---

### Official Review · Reviewer_DsUr · 2024-08-09
**Great novel contribution and results, but a few clarifications are necessary**

**Overall Rating:** 8
**Confidence:** 4

**Other Quality Metrics:**

(a) Clarity of writing: Excellent
(b) Clinical Significance: Great
(c) Methodological Novelty: Great
(d) Experiments and results: Great

**Questions For The Authors:**

* In section III.F, the first paragraph ends with "and", was this sentence supposed to continue to elaborate the PTA score?

* It is claimed that "an ideal model achieves a high PTA". Why is PTA a better metric than just performance on the new dataset? Why is catastrophic forgetting bad if the pretrained and finetuned model performs well on the second dataset?

**Strengths:**

* This paper addresses a major issue in the real implementation of machine learning in healthcare: lack of transferability between hospital systems due to differing distributions of clinical values. Specifically, they address a particularly challenged subset of this issue: transferability between large hospital systems to smaller ones.
* This benchmark leverages existing, openly accessible datasets. This is important for the applicability of this benchmark for others to easily evaluate their algorithms.
* The introduction does an excellent job of framing the issue, demonstrating DIL as an avenue for addressing it, and defining the scope of the paper.
* The methods are very clearly defined, to the extent that this work appears to be replicable from the paper alone, without needing to share any code.
* This paper very strongly supports the issue with a thorough demonstration of how distributions of clinical metrics vary across the hospital systems.
* While the new proposed DIL algorithm has the limitation of needing data to be transferred with the model, the authors accurately address this as a limitation, and further mention that one of the existing techniques (EWC) does not require any data to be transferred. Still, if I am not mistaken, the proposed approach does not require the entire original training set to be transferred with the model.

**Summary Of The Paper:**

Ultimately, the goal of this paper is to improve the extension of machine learning models from large hospital systems to smaller hospital systems, without losing the performance that the models demonstrate on the original population. This paper takes a step towards this goal by introducing a benchmark test for evaluating an algorithm's ability to adapt a model from a large dataset (MIMIC-III) to smaller regional datasets (in the eICU database), without losing the information it gained from the large dataset. The paper also demonstrates why this is historically problematic, since the distributions of values are not consistent across institutions/regions. The paper addresses this task as Domain Incremental Learning (DIL), and demonstrates how existing techniques for DIL can improve the model's generalizability. Additionally, the paper proposes a new DIL technique, which is a balanced combination of existing techniques, and demonstrates how it outperforms the existing techniques on many tasks.

**Weaknesses:**

* Throughout the paper, the use of the word "task" is confusing. It is used to refer to both the various prediction/classification tasks, and to the process of training on each dataset. Sometimes it is not clear which is being referred to. For example, in section III.F, the first sentence uses the word "task" twice, and I think that each of them is referring to the different usage of the word.
* The introduction to section III uses set notation to describe the structure of the two-stage training, but the use of the notation here feels contrived. I understand that this technically gives the algorithm flexibility to have more than 2 datasets, but this notation communicates that this is one big dataset, instead of two independent ones. This is particularly confusing given the above comment about the word "task". Also, this notation should include i ≠ j. Also, the sentence starts with "this section we will be be.." (small typo with double "be").
* The model used for the testing is a Bi-directional LSTM. The bi-directionality would mean that, unless specifically stopped from doing so, the model could use future information to make predictions. Training/evaluating the BiLSTM was unclear about whether the BiLSTM was only given past/present data for each time point prediction.

---

### Official Review · Reviewer_AJ9J · 2024-08-10
**A technically sound work with well designed experiments and potentially high clinical impact**

**Overall Rating:** 8
**Confidence:** 3

**Other Quality Metrics:**

- Clarity of writing: great
- Clinical significance: great
- Methodological novelty: good
- Experiments and results: great

**Questions For The Authors:**

- Although not explicitly stated, I assume that the violin plots imply that transferrability of the model across domains is mainly impaired by features showing a distribution shift. Do you think that this could be quantified in the model's performance, i.e. like a feature importance analysis?
- As a follow-up question, did you try removing these features that cause this distribution shift and applying the transfer learning for a reduced set of features? For example, by feature selection according to distribution differences; I'm thinking this could simplify the problem a lot, as some variables might not be that important for some tasks.
- Could the proposed method be applied in settings where only small hospitals are involved (small-to-small hospital transfer)? I understand that the proposed training objective requires a relatively high memory buffer size.

**Strengths:**

- The paper is well written and highlights the clinical significance of the task wonderfully.
- Problem statement and related work are adequately covered.
- The proposed benchmark is rich in data from heterogeneous sources. The pre-processing procedure is clearly presented and explained, I suggest that upon publication code is also shared for these steps to aid reproducibility. The benchmark tasks are adequate and relevant.
- The model (BiLSTM) is simple and adequate for highlighting the benchmark and the method.
- Technically, the experiments seem to be very sound (evaluation, tuning and selection).
- The violin plots present a very good visualization of the distribution shift across measurements in different regions.
- The results show the utility of the proposed benchmark, along with the potential of the proposed method.
- In general, the paper deals with a problem of great significance and interest. The proposed benchmark can have a large impact on the field, fostering further development of domain incremental methods.

**Summary Of The Paper:**

- The paper discusses the important topic of small hospital challenges in transfer learning settings, when high heterogeneity exists for data between different regions and hospitals.
- First, the paper proposes a formal benchmark for evaluating transfer learning algorithms, that utilizes two datasets (MIMIC-III and eICU) as hospital data sources, thus modeling a transfer learning setting from a big (MIMIC-III) to other small (eICU) hospitals. Four benchmark tasks are also proposed to evaluate performance of transfer learning algorithms.
- Then, the paper proposes a novel domain incremental loss, inspired by two existing methods (Data Replay and EWC) and designed to address their shortcomings.
- A BiLSTM model is trained for each task with cross-validation and hyperparameter selection.
- The presented results highlight the heterogeneity of measurements across different regions. The proposed method outperforms previous methods in most of the benchmark tasks.

**Weaknesses:**

- As the paper mentions, it is hard to understand the contribution of the proposed loss, as it is designed to address multiple tasks, although here only two are shown.
- There are some parts of the paper that are hard to follow, e.g. page 5, 2nd last paragraph from the end of the left column, paragraph ends with "and".

---

### Decision · Program_Chairs · 2024-09-22

Accept